# Performance bonuses and the quality of primary health care delivered by family health teams in Brazil: A difference-in-differences analysis

**Nasser Fardousi**[1], **Everton Nunes da Silva**[2], **Roxanne Kovacs**[1], **Josephine Borghi**[1], **Jorge O. M. Barreto**[3], **Søren Rud Kristensen**[4], **Juliana Sampaio**[5], **Helena Eri Shimizu**[2], **Luciano B. Gomes**[5], **Letícia Xander Russo**[6], **Garibaldi D. Gurgel**[7], **Timothy Powell-Jackson**[1] *

**1** Department of Global Health and Development, London School of Hygiene & Tropical Medicine, London, United Kingdom, **2** University of Brasília, Brasília, Brazil, **3** Oswaldo Cruz Foundation-Fiocruz, Brasília, Brazil, **4** Centre for Health Economics, University of Southern Denmark, Odense, Denmark, **5** Department of Health Promotion, Federal University of Paraíba, João Pessoa, Paraíba, Brazil, **6** Federal University of Grande Dourados, Mato Grosso do Sul, Brazil, **7** Oswaldo Cruz Foundation-Fiocruz, Pernambuco, Brazil

* timothy.powell-jackson@lshtm.ac.uk

**Data Availability Statement:** The PMAQ scores for each family health team (our measure of quality of care) and the responses from a survey of

## Abstract

### Background

Pay-for-performance (P4P) programmes to incentivise health providers to improve quality of care have been widely implemented globally. Despite intuitive appeal, evidence on the effectiveness of P4P is mixed, potentially due to differences in how schemes are designed. We exploited municipality variation in the design features of Brazil's National Programme for Improving Primary Care Access and Quality (PMAQ) to examine whether performance bonuses given to family health team workers were associated with changes in the quality of care and whether the size of bonus mattered.

### Methods and findings

For this quasi-experimental study, we used a difference-in-differences approach combined with matching. We compared changes over time in the quality of care delivered by family health teams between (bonus) municipalities that chose to use some or all of the PMAQ money to provide performance-related bonuses to team workers with (nonbonus) municipalities that invested the funds using traditional input-based budgets. The primary outcome was the PMAQ score, a quality of care index on a scale of 0 to 100, based on several hundred indicators (ranging from 598 to 660) of health care delivery. We did one-to-one matching of bonus municipalities to nonbonus municipalities based on baseline demographic and economic characteristics. On the matched sample, we used ordinary least squares regression to estimate the association of any bonus and size of bonus with the prepost change over time (between November 2011 and October 2015) in the PMAQ score. We performed subgroup analyses with respect to the local area income of the family health team. The

municipality managers (our exposure variables) were provided to the authors through a collaborative agreement with the Department for Family Health at the Ministry of Health of Brazil. Requests for access to these data should be directed at the Department for Family Health: telephone number +55 61 33159044 or email desf@saude.gov.br. All other data used are publicly available at https://doi.org/10.17037/DATA. 00002886.

**Funding:** This research was funded by the Medical Research Council, Newton Fund and the Brazilian National Council for the States Funding Agencies (CONFAP) under the UK to Brazil Joint Health Systems Research Call (grant MR/R022828/1). The MRC grant was awarded to JB and TPJ. Funding from CONFAP came from Fundação de Amparo à Pesquisa do Distrito Federal (FAPDF), Fundação de Amparo à Ciência e Tecnologia do Estado de Pernambuco (FACEPE) and Fundação de Apoio à Pesquisa do Estado da Paraíba (FAPESQ). CONFAP funding was awarded to ES. The funders had no role in study design, data collection and analysis, decision to publish, or preparation of the manuscript.

**Competing interests:** The authors have declared that no competing interests exist.

**Abbreviations:** GDP, gross domestic product; OECD, Organisation for Economic Co-operation and Development; PMAQ, Programme for Improving Primary Care Access and Quality; P4P, pay-for-performance; STROBE, Strengthening the Reporting of Observational Studies in Epidemiology.

matched analytical sample comprised 2,346 municipalities (1,173 nonbonus municipalities; 1,173 bonus municipalities), containing 10,275 family health teams that participated in PMAQ from the outset. Bonus municipalities were associated with a 4.6 (95% CI: 2.7 to 6.4; $p < 0.001$) percentage point increase in the PMAQ score compared with nonbonus municipalities. The association with quality of care increased with the size of bonus: the largest bonus group saw an improvement of 8.2 percentage points (95% CI: 6.2 to 10.2; $p < 0.001$) compared with the control. The subgroup analysis showed that the observed improvement in performance was most pronounced in the poorest two-fifths of localities. The limitations of the study include the potential for bias from unmeasured time-varying confounding and the fact that the PMAQ score has not been validated as a measure of quality of care.

## Conclusions

Performance bonuses to family health team workers compared with traditional input-based budgets were associated with an improvement in the quality of care.

## Author summary

### Why was this study done?

- Pay-for-performance (P4P) programmes to incentivise health providers to improve quality of care have been widely implemented globally.

- P4P schemes vary considerably in how they are designed, but there is limited evidence on whether these design choices matter for quality of care.

- The National Programme for Improving Primary Care Access and Quality (PMAQ) in Brazil was a P4P scheme that gave municipalities autonomy to decide how funds could be spent and, specifically, whether payments could be used to reward health workers.

### What did the researchers do and find?

- We used a difference-in-differences approach to examine whether performance bonuses given to family health team workers, compared with traditional input-based budgets, were associated with changes in the quality of care, and whether the size of bonus mattered.

- We found that giving bonuses to family health team workers was associated with an improvement in the quality of care, and the association increased with the size of bonus.

- Improvements in quality of care were most pronounced for family health teams located in the poorest two-fifths of areas.

**What do these findings mean?**

- The findings suggest that performance bonuses to family health team workers can potentially be a more effective way of using PMAQ funds to improve quality of care than input-based budgeting.

- Performance bonuses to family health team workers appeared to reduce inequalities in the delivery of primary health care.

- Further research is needed to better understand what other design features, such as who gets paid and the frequency of payment, influence the extent to which P4P schemes improve quality of care.

Please see *S1 Portuguese Abstract* for an alternate language Abstract.

## Introduction

Primary health care is the foundation of many health systems. The vital role it plays, as a stepping stone towards achieving universal health coverage, is widely recognised [1]. Over the past two decades, Brazil has implemented sweeping primary health care reforms, of which the most high profile and consequential component was the Family Health Strategy [2,3]. According to this policy, family health teams spearhead primary health service provision at the community level free of charge [3]. Through public financing, family health teams were rapidly scaled up in communities across the country, resulting in improvements in population health [4–7]. Nonetheless, concerns over quality of care have persisted [8,9]. In 2011, Brazil introduced a national health financing programme to improve access to and quality of primary health care. Under this National Programme for Improving Primary Care Access and Quality (*Programa Nacional de Melhoria do Acesso e da Qualidade da Atenção Básica* [PMAQ]), the federal government made financial payments to municipalities based on the performance of family health teams.

Pay-for-performance (P4P) has been widely applied in the United States, the United Kingdom, other Organisation for Economic Co-operation and Development (OECD) countries, and increasingly in low- and middle-income countries [10–13]. The idea of linking financial payments to the performance of health providers has intuitive appeal. In practice, however, results from empirical studies are mixed, and key research questions remain unanswered, making it difficult to draw firm conclusions [14]. A possible explanation for the mixed results is that the specific design of P4P schemes can vary on many dimensions [15–17] and, although these design elements are likely of key importance for effectiveness, the evidence base for making informed choices is lacking [18].

PMAQ provides an ideal testing ground for addressing three questions about scheme design for which there is limited evidence. First, a key design decision in P4P schemes concerns how the money can be spent and, specifically, whether payments should be used to reward health workers. As a federal programme, many of the design features of PMAQ were national in scope. However, the programme was required to give municipalities discretion on how funds could be used, which means we can compare municipalities that gave performance bonuses to health workers with those that invested the funds entirely through traditional input-based budgets. Second, municipalities differed in the size of bonus paid to family health

team workers. Intuitively and theoretically, the size of incentive should matter for health service delivery and performance [19]. However, there are only a few studies in high-income countries that have examined the effect of bonus size [20,21]. It remains an open and pertinent question in low- and middle-income countries where health sector resources are more constrained [22,23]. Finally, because municipalities had the flexibility to decide whether to retain payments at the municipal level or redirect them to the family health team level, we can also speak to the question of whether varying the level of payment matters. Economic theory suggests that payments made closer to the executing level are more likely to be effective due to a reduced risk of free riding [24] and empirical evidence from one high-income country supports this hypothesis [25].

We examined whether performance bonuses given to family health team workers were associated with changes in the quality of care and whether the size of bonus mattered in Brazil. While PMAQ has been previously associated with reduced socioeconomic inequality in performance across teams, no study thus far has evaluated the impact of variations in PMAQ's design features across municipalities [26]. We hypothesised that performance bonuses to health workers provide a stronger incentive to improve the quality of primary health care than traditional input-based financing. Using national programme data, our primary analysis focused on family health teams providing care to approximately 35.4 million people.

## Methods

### Study setting and design

Family health teams are the lynchpin of the primary health care system in Brazil. They are the first point of contact for the community, providing primary health care to a catchment population of approximately 3,500 people. Each family health team operates from a health facility and comprises at least one physician, nurse, nurse assistant, full-time community health worker, and in some teams, dentist and oral health staff. PMAQ is described in more detail elsewhere [26]. In brief, it was a national programme that allocated around 10% of federal primary health care funds to municipalities based on the performance of family health teams [27]. It was implemented over three rounds between 2011 and 2019. At the beginning of each round, the performance of family health teams was assessed through a combination of self-assessment, routine monitoring, and independent external evaluation that was led by universities.

The assessment involved measurement of hundreds of indicators (598 in round 1 and 660 in round 3), some of which changed across the different rounds [28–30]. Indicators included those relating to service availability (e.g., opening hours), structural quality of care (e.g., availability of medicines), processes of care (e.g., content of care and treatment completion), outcomes (e.g., patient satisfaction and birth weight of children), utilisation of health care (e.g., patient volume), and management (e.g., appointments scheduled). Of the external evaluation indicators ($n$ = 648) used in the third round of PMAQ, the most common were measures of structural quality (58.5%), followed by management practices (10.9%), clinical processes of care (10.7%), service availability (8.7%), outcome (8.3%), and utilisation (0.9%), with 2.0% unclassified.

Achievement of targets linked to each indicator was used to generate a summary measure of performance, known as the PMAQ score [28–30]. To calculate the score, the number of points achieved was divided by the number of points available in each of the three indicator categories. A weighted average across the categories was then multiplied by 100 to give the PMAQ score. The weights given to each indicator category changed between rounds, with slightly more weight given to routine monitoring indicators in round 3, at the expense of external evaluation indicators [26]. On the basis of this score, each participating family health team

was placed into a performance group that determined the monthly financial reward for the entire implementation round. The amount of money each municipality received was the sum of the specific rewards of family health teams within the municipality. A key feature of the federal design was that the performance groups were determined by the relative performance of family health teams within socioeconomic bands in the first two rounds of PMAQ. In round 3, performance groups were based simply on absolute PMAQ scores, with no adjustment for socioeconomic inequality.

Our study exploited the fact that municipalities, as the decentralised administrative health authority in Brazil, had autonomy in how PMAQ funds could be spent. Some municipalities chose to use some or all of the money to provide performance-related bonuses (henceforth bonus municipalities) to supplement the income of family health team members. In other words, financial incentives were passed down to the health provider level, with potential implications for worker motivation. In our sample, the majority of bonus municipalities (79.6%) stated that the bonuses were linked to performance on the external evaluation, and almost all (96.7%) gave the bonuses to every member of the family health team. The nonbonus municipalities spent the PMAQ funds in the traditional way using input-based budgets to purchase drugs and equipment and support infrastructure, training and management. This potentially improved facility readiness to provide care and the conditions of work but not the remuneration of health workers.

Our study design compared changes over time in quality of care between bonus and nonbonus municipalities. Specifically, we used a difference-in-differences approach combined with matching, a study design that has been shown to perform well in limiting bias [31] and has been widely applied in the evaluation of health policies. The matching sought to improve baseline balance between municipalities that passed on bonuses to family health team workers and those that did not give bonuses. Matching on pretreatment outcomes is attractive as it can improve balance for unobserved time-varying confounders [32,33]. The difference-in-differences method then controls for unobserved but fixed omitted variables, relying on the assumption that the counterfactual trends in the treatment and control groups are the same. The matching procedure makes this assumption of parallel trends more plausible by ensuring that the outcomes in the treatment and control groups are similar in levels at baseline [34–36]. Another useful property of matching is that it reduces bias from the potential misspecification of the subsequent regression model [37].

The study received ethics approval from the University of Brasilia (Brasilia, Brazil; CAAE 30424620.4.0000.8093), and the London School of Hygiene & Tropical Medicine (London, UK; 15805). The analysis in this study was planned in June 2018. A prospective study protocol or analysis plan is not available. This study is reported as per the Strengthening the Reporting of Observational Studies in Epidemiology (STROBE) guideline (S1 Checklist).

## Data sources

We used data from five sources. First, we obtained the PMAQ score and performance category of all family health teams in each implementation round from the Ministry of Health. Second, to capture variation in the design of PMAQ, we used data from an online survey of municipality health managers, conducted as part of the external evaluation in the third round of implementation. This survey asked various questions on incentive design, including whether the municipality passed on PMAQ funds as bonuses to family health team workers and the size of the bonuses as a percentage of staff salaries. Third, we used the 2010 Brazilian Population Census to measure the average monthly income of households in each census area. We geographically linked each health facility to a census sector, allowing us to measure the local area income

of each family health team [26]. Fourth, we obtained data on the characteristics of health facilities to which family health teams were attached from a census of health facilities done by the Ministry of Health in 2011. Fifth, we used established sources to construct a dataset of municipality socioeconomic and demographic characteristics for the year 2010 (S1 Table).

## Measures

Our primary outcome was the PMAQ score, which we regard as a broad measure of quality of care. It was a composite measure based on indicators of service availability, structural quality of care, processes of care, outcomes, utilisation of healthcare, and management. The score, calculated by the Ministry of Health, could range from 0 (lowest possible score) to 100 (highest possible score) and was interpreted as the percentage of the maximum score obtainable by a family health team. The PMAQ score in round 3 was based on measurement of performance around October 2015. The PMAQ score in round 1 reflected performance around November 2011, at the beginning of the programme, before PMAQ funds had been disbursed. It therefore acted as a baseline.

The exposure variables were (i) whether the municipality used PMAQ funds to provide bonuses to family health team members; or (ii) categories indicating the size of bonus as a proportion of staff salaries (1% to 20%; 21% to 50%; more than 50%), with 0 % as the reference category. Some municipalities reported that the size of bonus they gave to teams varied and hence could not be categorised. These municipalities were dropped from the analysis of bonus size. Covariates included municipality characteristics (gross domestic product [GDP] per capita, human development index, Gini index, population, urban share of population, share of population under 5 years, share of population over 60 years, and average monthly PMAQ funds awarded per team in round 1), facility characteristics (type of health facility and number of clinical staff), and local area characteristics (monthly income per capita).

## Statistical analyses

We used a difference-in-differences approach to examine the association between performance bonuses and quality of care. We analysed the data at the team level, creating a panel of teams that took part in the first and third round of PMAQ. Our regression models estimated the prepost change over time (the difference between round 3 and round 1) in the PMAQ score of family health teams in municipalities that gave bonuses relative to comparison municipalities that did not give bonuses. To explore whether there was a dose–response relationship, we replaced the binary exposure variable with dummy variables indicating the size of bonus. We fitted ordinary least squares regression. Robust standard errors were clustered at the municipality level given that exposure to bonuses varied at this level, as is standard in the health policy evaluation literature [33].

We controlled for the aforementioned covariates. One particular concern was that municipalities which gave bonuses may have also received more funding from PMAQ than those that did not [38]. By including the initial amount of PMAQ funding per team awarded to municipalities—an amount determined in the round 1 assessment at the beginning of the programme—we sought to deal with this potential source of confounding, thereby separating the incentive effect of bonuses from the influence of simply more financial resources. Because the models were estimated in first differences, the inclusion of the covariates meant we controlled for differential trends in quality based on initial values of these variables.

The key assumption underpinning any difference-in-differences approach is that the counterfactual outcomes for the treatment and comparison groups follow the same trend [39]. To increase the plausibility of the parallel trend assumption, we used propensity score methods to create a comparison group of nonbonus municipalities that best matched the bonus

municipalities at baseline. We estimated propensity scores with probit regression using GDP per capita, human development index, Gini index, population, urban share of population, share of population under 5 years, share of population over 60 years, and the mean PMAQ score at baseline as predictors. We then performed one-to-one matching of municipalities, with no replacement and a calliper of 0.01. Bonus municipalities that could not be matched to a comparison municipality and nonbonus municipalities that were not the nearest match to a bonus municipality were discarded. To evaluate the matching procedure, we compared baseline balance between bonus and nonbonus municipalities and reported the *p*-value from a *t* test of the difference (see also S2 and S3 Tables and S1 Fig). We report throughout results from a difference-in-differences approach without matching, in light of evidence suggesting that matching can introduce regression to the mean bias under certain restrictive conditions [35,40].

We conducted subgroup analyses to examine whether the association between bonuses and quality differed by the local area income of where family health teams were located. We categorised family health teams into five groups of equal size by local area income and included an interaction between this variable and treatment status (any bonus or size of bonus) in the main estimating equation. Using the margins command in Stata, we report the absolute effect for each subgroup, as well as the effect relative to the mean PMAQ score in round 1.

We performed several sensitivity analyses. First, the indicators and formula used to generate the PMAQ score varied across rounds. Although the difference-in-differences approach in principle deals with the change in measurement, the potential for bias remains. Based on the suggestion of a reviewer, we developed a structural quality of care index using a common set of 123 indicators from each PMAQ round that captured the availability of drugs, equipment, consumables, and diagnostic tests. We defined this index as the percentage of items available in each facility during the external assessment visit. We examined the sensitivity of our main findings to this alternative measure of performance. Second, we included additional control variables in the regression models. Third, we used a lagged dependent variable model as an alternative approach, since it is based on a different identifying assumption of conditional independence [32,39]. Fourth, we experimented with different callipers in the matching procedure. Fifth, we produced results for municipalities that gave bonuses to health workers but stated that the amount was not fixed. All analyses were done in Stata 16.1 SE.

## Results

Of the 5,570 municipalities in Brazil, 5,028 (90.2%) implemented PMAQ and provided information on whether they gave bonuses to family health team members in round 3 (S2 Fig). We excluded 1,585 (31.5%) of 5,028 municipalities that had no family health team participating in both round 1 and round 3 of the programme and a further 72 (1.4%) municipalities that had no family health team with complete data on local area income and facility characteristics. Our unmatched analytical sample comprised 3,371 municipalities (1,937 nonbonus municipalities; 1,434 bonus municipalities), containing 13,716 family health teams (7,575 teams in nonbonus municipalities; 6,141 teams in bonus municipalities). After matching, the analytical sample comprised 2,346 municipalities (1,173 nonbonus municipalities; 1,173 bonus municipalities), containing 10,275 family health teams (5,052 teams in nonbonus municipalities; 5,223 teams in bonus municipalities). For the analysis of bonus size (S3 Fig), we excluded from the unmatched sample 610 municipalities whose size of bonus could not be categorised because it was reported to vary, leaving an unmatched analytical sample of 2,761 municipalities (1,937 with 0% bonus; 332 with 1% to 20% bonus; 359 with 21% to 50% bonus; 133 with >50% bonus) containing 11,060 family health teams (7,575 with 0% bonus; 1,197 with 1% to 20% bonus; 1,467 with 21% to 50% bonus; 821 with >50% bonus).

**Table 1. Baseline characteristics of sample.**

| | Full (unmatched) sample | | | Matched sample | | |
|---|---|---|---|---|---|---|
| | Any bonus to family health teams | No bonus to family health teams | *p*-Value | Any bonus to family health teams | No bonus to family health teams | *p*-Value |
| *Family health teams and local area* | | | | | | |
| Number of family health teams | 6,141 | 7,575 | | 5,223 | 5,052 | |
| PMAQ score round 1 | 61.4 (9.3) | 60.7 (10.4) | 0.514 | 61.3 (9.5) | 61.5 (10.1) | 0.879 |
| Local area monthly income per capita | 1.40 (0.82) | 1.69 (0.88) | <0.001 | 1.47 (0.84) | 1.65 (0.95) | 0.043 |
| *Health facilities* | | | | | | |
| Number of facilities | 5,523 | 5,988 | | 4,626 | 3,859 | |
| Facility type | | | | | | |
| Health centre | 4,189 (75.85%) | 4,479 (74.8%) | 0.480 | 3,590 (77.6%) | 2,806 (72.7%) | 0.008 |
| Health post and other | 1,334 (24.2%) | 1,509 (25.2%) | | 1,036 (22.4%) | 1,053 (27.3%) | |
| Number of clinical staff | 14.47 (6.83) | 17.22 (11.46) | 0.012 | 14.64 (7.1) | 17.97 (12.78) | 0.039 |
| *Municipalities* | | | | | | |
| Number of municipalities | 1,434 | 1,937 | | 1,173 | 1,173 | |
| PMAQ funds per FHT in round 1 | 4,849 (2141) | 4,449 (2190) | <0.001 | 4,718 (2,150) | 4,639 (2,150) | 0.375 |
| GDP per capita | 10.86 (12.91) | 14.13 (15.46) | <0.001 | 11.99 (13.93) | 11.89 (11.82) | 0.840 |
| Human development index | 0.65 (0.07) | 0.68 (0.07) | <0.001 | 0.66 (0.07) | 0.65 (0.07) | 0.546 |
| Gini index | 0.51 (0.06) | 0.49 (0.07) | <0.001 | 0.50 (0.06) | 0.51 (0.06) | 0.484 |
| Total population | 0.35 (1.04) | 0.47 (3.21) | 0.172 | 0.39 (1.14) | 0.49 (3.92) | 0.371 |
| Share of population urban | 0.64 (0.22) | 0.66 (0.22) | 0.001 | 0.65 (0.22) | 0.64 (0.21) | 0.717 |
| Share of population under 5 years | 0.07 (0.01) | 0.07 (0.02) | <0.001 | 0.07 (0.01) | 0.07 (0.01) | 0.869 |
| Share of population over 60 years | 0.12 (0.03) | 0.12 (0.03) | 0.251 | 0.12 (0.03) | 0.12 (0.03) | 0.706 |

Data are *n* (%) or mean (SD). PMAQ score is an index of quality between 0 and 100. Local area monthly income per capita is in Brazilian real divided by 1,000. Total population is divided by 100,000.

FHT, family health team; GDP, gross domestic product; PMAQ, National Programme for Improving Primary Care Access and Quality.

Table 1 presents descriptive statistics at baseline for bonus municipalities and nonbonus municipalities. In the full sample without matching, the mean PMAQ score was similar between treatment and control municipalities. However, the bonus municipalities received significantly more PMAQ funds per team and had lower GDP per capita and human development, greater income inequality, lower share of the population living in urban areas, and higher share of the population under the age of 5 years. By contrast, in the matched sample, there were no statistically significant differences between treatment and control for municipality characteristics, and the PMAQ score in round 1 was almost identical, indicating that the matching procedure achieved good balance.

Figs 1 and 2 show the mean PMAQ score in round 1 and round 3 in bonus and nonbonus municipalities as well as those categorised by size of bonus (see also S4 and S5 Figs). In the matched sample, the change over time in the mean PMAQ score was −1.65 points in control municipalities and 2.69 points in bonus municipalities, representing an unadjusted difference between the two groups of municipalities (difference-in-differences) of 4.3 points (95% CI 1.7 to 6.9; *p* = 0.001). The difference between control and the size of bonus groups of municipalities in the change over time in the mean PMAQ score was: 2.6 points (95% CI −1.6 to 6.7;

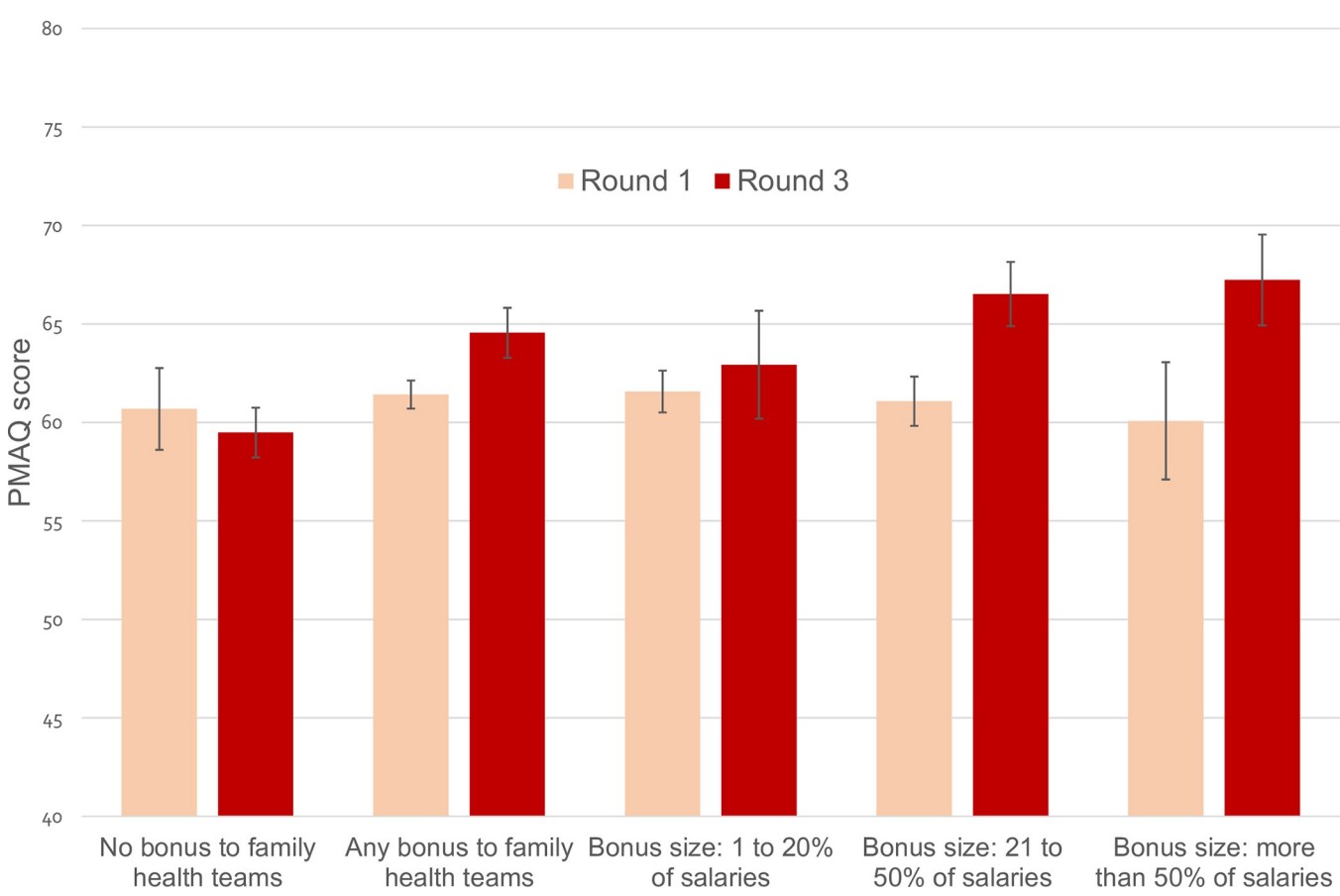

**Fig 1. PMAQ score by municipality bonus design in the unmatched sample.** PMAQ, National Programme for Improving Primary Care Access and Quality.

$p = 0.231$) for the small (1% to 20%) bonus size group; 6.7 points (95% CI 3.8 to 9.7; $p < 0.001$) for the medium (21% to 50%) size group; and 8.6 points (95% CI 5.5 to 11.7; $p < 0.001$) for the large (more than 50%) size group. The unadjusted difference-in-differences estimates on quality of care were similar, if a little larger, in the full sample without matching.

Table 2 presents the difference-in-differences regression estimates of the association between performance bonuses and quality of care. The results from the matched analysis show that the change over time in the PMAQ score was 4.6 points (95% CI: 2.7 to 6.4; $p < 0.001$) greater in the bonus municipalities compared with the nonbonus municipalities. This association was equivalent to a relative increase of 7.5% (over the baseline mean of 61.4 in the PMAQ score). The magnitude of the associations increased with the size of bonus, suggesting a dose–response relationship. The change over time in the PMAQ score was 8.2 points (95% CI: 6.2 to 10.2; $p < 0.001$) greater in the municipalities giving the largest bonuses compared with the nonbonus municipalities. The results from the analysis without matching were similar.

Fig 3 presents the subgroup effects with respect to local area income, revealing several patterns in the data (see also S4 Table). First, the association between bonuses and quality of care was U-shaped across the income distribution—that is, changes in quality of care were largest for teams in the poorest localities, fell as income of the catchment area increased, and then rose again in the richest quintile. Second, these differences were most pronounced when small bonuses were given, such that small bonuses were associated with better quality of care only for teams in the poorest two-fifths of areas. Third, large bonuses were associated with better

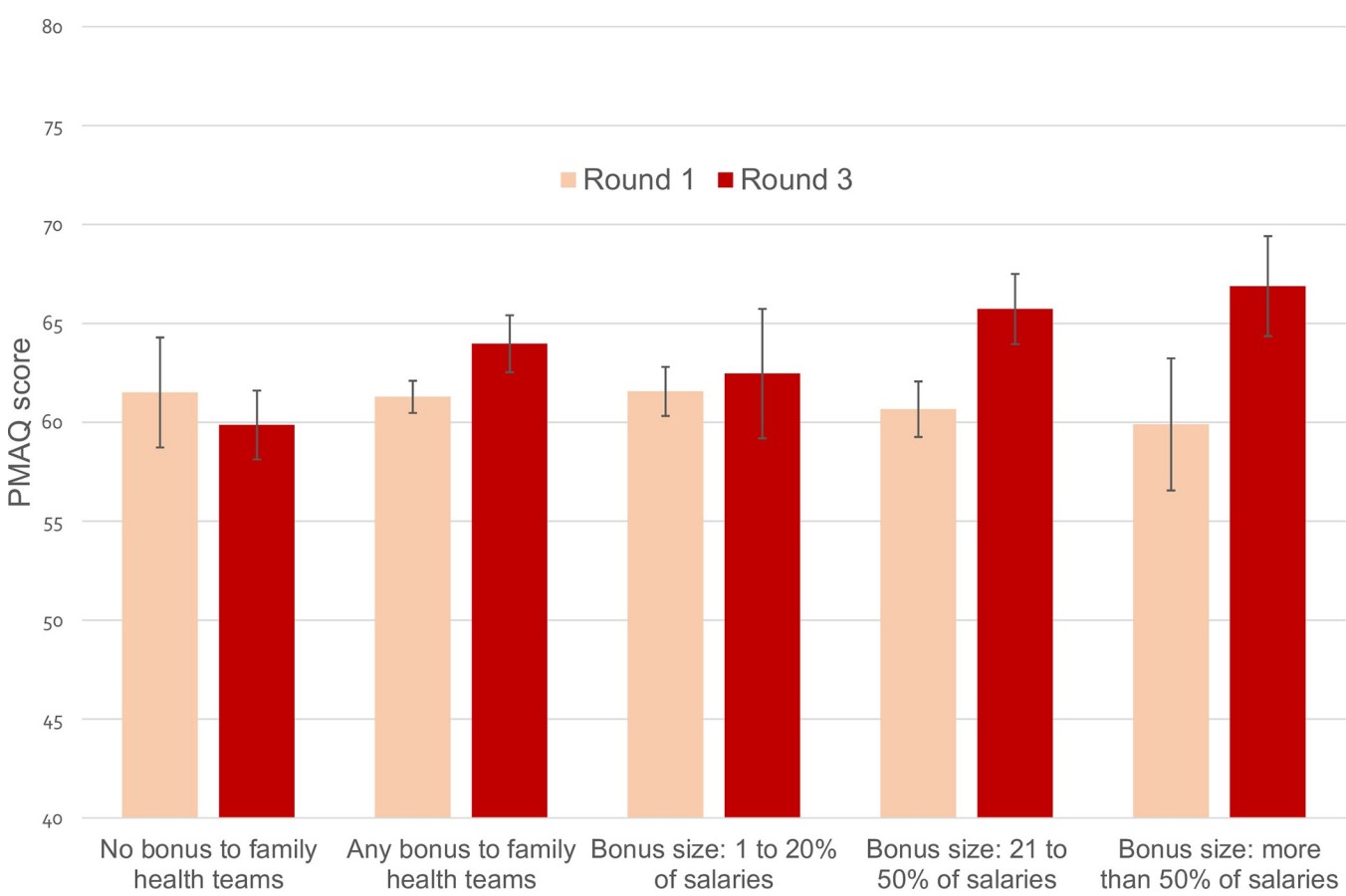

**Fig 2. PMAQ score by municipality bonus design in the matched sample.** PMAQ, National Programme for Improving Primary Care Access and Quality.

quality of care in teams across the income distribution and heterogeneity between income groups was less pronounced (S4 Table). Taken together, the results indicate that changes in quality were largest for teams in the poorest two-fifths of localities, but the size of bonus mattered most for teams in the richest three-fifths of localities.

We performed several sensitivity analyses. The pattern of results was similar when we used our structural quality of care index, although associations were smaller in magnitude (S5 Table). Results were also similar when we used a lagged dependent variable model in which we regressed the PMAQ score in round 3 on the incentive design indicator(s), baseline covariates, and the PMAQ score in round 1 (S6 Table). It is also worth noting that the coefficient on the initial amount of PMAQ funding per team awarded to municipalities was positive, implying a positive resource effect on quality. The results were generally not sensitive to the inclusion of additional controls, or the calliper value used in the matching. In some specifications, the difference-in-differences estimate for small (1% to 20%) bonuses was no longer significant but otherwise the findings were similar (S7 Table). We also report the results for municipalities who gave bonuses to health workers but stated that the amount was not fixed, with estimates slightly larger than the small (1% to 20%) bonus group.

## Discussion

Our study examined the relationship between bonus payments to frontline primary health care workers and quality of care by exploiting variation in how municipalities decided to use

**Table 2. Association between bonuses to family health team workers and the PMAQ score in the full (unmatched) and matched samples of municipalities.**

| | Full (unmatched) sample | | | | Matched sample | | | |
|---|---|---|---|---|---|---|---|---|
| | Any bonus to family health teams | | Size of bonuses | | Any bonus to family health teams | | Size of bonuses | |
| | Coefficient (95% CI) | p-Value | Coefficient (95% CI) | p-Value | Coefficient (95% CI) | p-Value | Coefficient (95% CI) | p-Value |
| **PMAQ bonus** | | | | | | | | |
| Municipalities giving bonuses | 4.2 (2.6 to 5.7) | <0.001 | | | 4.6 (2.7 to 6.4) | <0.001 | | |
| **PMAQ bonus size** | | | | | | | | |
| 1 to 20% of salaries | | | 2.6 (−0.4 to 5.6) | 0.0841 | | | 3.1 (−0.1 to 6.4) | 0.0609 |
| 21 to 50% of salaries | | | 6.2 (4.3 to 8.1) | <0.001 | | | 6.4 (4.1 to 8.6) | <0.001 |
| More than 50% of salaries | | | 7.5 (5.6 to 9.3) | <0.001 | | | 8.2 (6.2 to 10.2) | <0.001 |
| **Local area** | | | | | | | | |
| Poorer | −1.5 (−2.4 to −0.7) | <0.001 | −1.4 (−2.3 to −0.5) | 0.0029 | −1.6 (−2.5 to −0.6) | 0.0013 | −1.5 (−2.6 to −0.5) | 0.0039 |
| Middle | −1.2 (−2.2 to −0.2) | 0.0146 | −1.1 (−2.2 to 0.0) | 0.0580 | −1.1 (−2.3 to 0.0) | 0.0608 | −0.8 (−2.1 to 0.5) | 0.2061 |
| Richer | −1.7 (−2.8 to −0.6) | 0.0028 | −1.8 (−2.9 to −0.6) | 0.0028 | −1.8 (−3.2 to −0.4) | 0.0143 | −1.9 (−3.4 to −0.5) | 0.0091 |
| Richest | −1.0 (−2.3 to 0.3) | 0.1467 | −1.4 (−2.6 to −0.2) | 0.0226 | −0.9 (−2.4 to 0.5) | 0.2173 | −1.5 (−2.8 to −0.2) | 0.0275 |
| **Health facility** | | | | | | | | |
| Health centre | 0.1 (−0.6 to 0.9) | 0.7152 | 0.6 (−0.2 to 1.4) | 0.1694 | 0.1 (−0.8 to 0.9) | 0.8717 | 0.6 (−0.3 to 1.5) | 0.2074 |
| Number of clinical staff | -0.0 (−0.1 to 0.0) | 0.3863 | −0.0 (−0.1 to 0.0) | 0.4906 | 0.0 (−0.0 to 0.0) | 0.9160 | 0.0 (−0.0 to 0.0) | 0.7443 |
| **Municipality characteristics** | | | | | | | | |
| PMAQ funds in round 1 (in R$ 1,000) | −2.3 (−2.6 to −2.0) | <0.001 | −2.3 (−2.6 to −1.9) | <0.001 | −2.4 (−2.8 to −2.0) | <0.001 | −2.3 (−2.7 to −2.0) | <0.001 |
| GDP per capita (in R$ 1,000) | −0.1 (−0.1 to −0.0) | 0.0208 | −0.0 (−0.1 to 0.0) | 0.1108 | −0.1 (−0.1 to 0.0) | 0.1055 | −0.0 (−0.1 to 0.1) | 0.9148 |
| Human development index | −13.7 (−28.8 to 1.4) | 0.0758 | −8.4 (−24.6 to 7.8) | 0.3103 | −14.7 (−31.8 to 2.5) | 0.0936 | −17.5 (−34.6 to −0.4) | 0.0443 |
| Gini index | −5.2 (−29.8 to 19.5) | 0.6819 | 8.2 (−4.5 to 20.8) | 0.2060 | −12.9 (−44.6 to 18.8) | 0.4249 | 3.5 (−12.4 to 19.4) | 0.6668 |
| Total population | 0.0 (−0.0 to 0.1) | 0.0964 | 0.0 (−0.0 to 0.0) | 0.4417 | 0.0 (−0.0 to 0.1) | 0.0671 | 0.0 (−0.0 to 0.1) | 0.4060 |
| Share of population urban | −2.9 (−6.3 to 0.6) | 0.1009 | −3.8 (−7.6 to −0.0) | 0.0472 | −2.1 (−6.3 to 2.1) | 0.3260 | −2.2 (−6.7 to 2.4) | 0.3556 |
| Share of population under 5 years | 54.3 (−49.1 to 157.6) | 0.3031 | 36.4 (−52.2 to 124.9) | 0.4207 | 94.9 (−35.1 to 224.8) | 0.1526 | 68.1 (−36.7 to 172.9) | 0.2025 |
| Share of population over 60 years | 33.0 (−10.0 to 75.9) | 0.1325 | 36.1 (−12.4 to 84.6) | 0.1441 | 60.7 (10.0 to 111.4) | 0.0190 | 72.5 (15.8 to 129.2) | 0.0122 |
| N teams | 13,716 | | 11,060 | | 10,275 | | 7,938 | |
| N municipalities | 3,371 | | 2,761 | | 2,346 | | 1,836 | |
| R-squared | 0.171 | | 0.186 | | 0.177 | | 0.200 | |

The dependent variable is the change in the PMAQ score, which is an index of quality between 0 and 100. The reference groups are as follows: for PMAQ bonus is nonbonus municipalities; for PMAQ bonus size is nonbonus municipalities; for local area is poorest; and for health centre is health post and others.

CI, confidence interval; FHT, family health team; GDP, gross domestic product; PMAQ, National Programme for Improving Primary Care Access and Quality.

funds under PMAQ. We found that giving bonuses to workers was associated with a significant increase in quality of care as measured by the PMAQ score. Improvements in quality of care were most pronounced for family health teams located in the poorest two-fifths of areas. The association with quality of care increased with size of bonus, suggesting a dose–response relationship. It is important to emphasise that we did not evaluate PMAQ, and the results should not be interpreted as estimates of the impact of the programme.

Compared with the control group, the PMAQ score in bonus municipalities increased by 4.6 points, equivalent to a relative increase of 7.5%. It is difficult to directly compare findings across studies because of the wide range of quality of care outcomes used in the literature. Our study connects most closely to the P4P literature in which studies have compared P4P with equivalent levels of input-based funding to disentangle the resource effect from the incentive

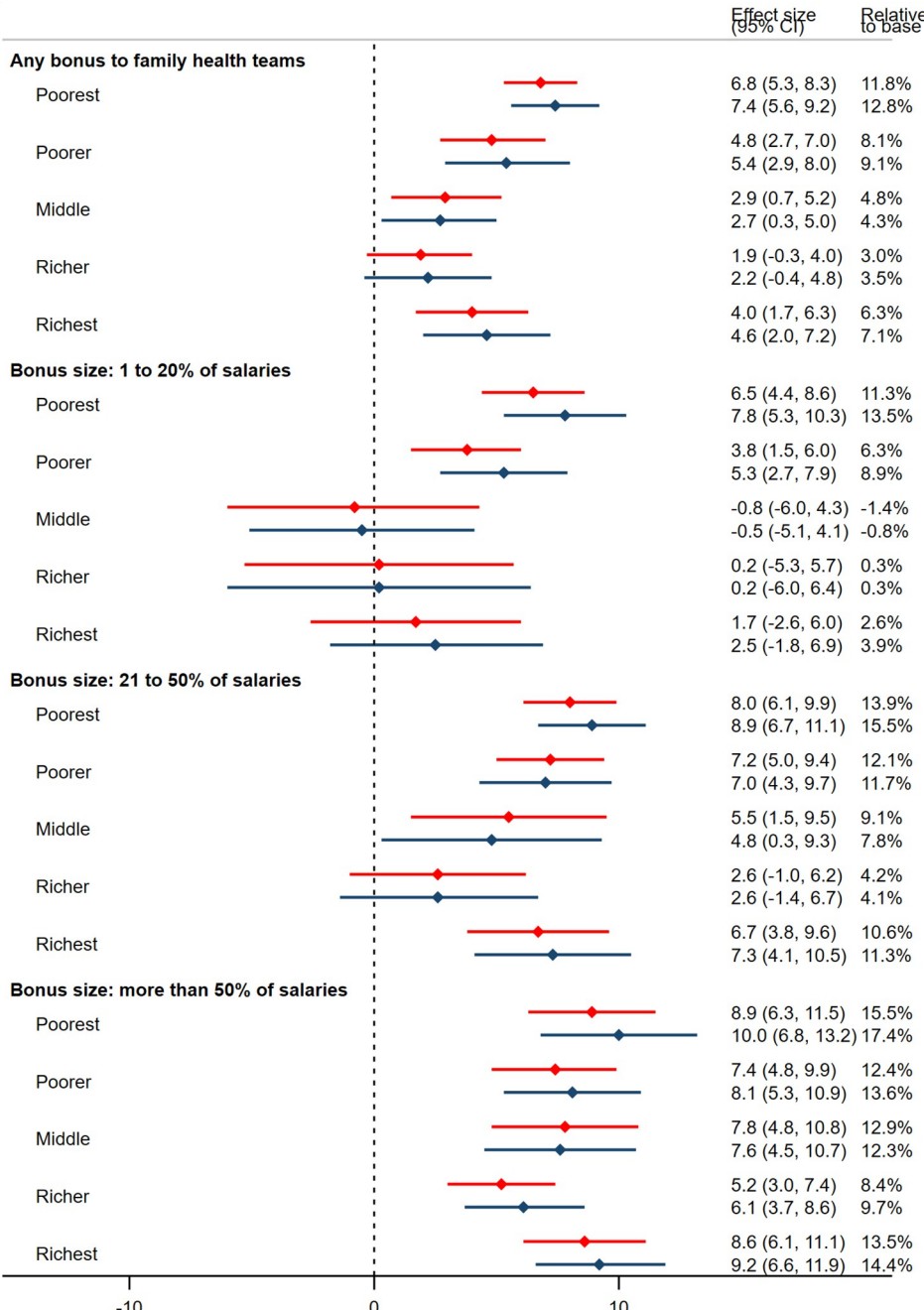

**Fig 3. Income subgroup analyses.** Red bars represent point estimates and 95% confidence intervals in the unmatched sample; blue bars represent point estimates and 95% confidence intervals in the matched sample. Income subgroups were defined using the average monthly income of households in the local area of a family health team. CI, confidence interval.

effect. In Rwanda, P4P increased tetanus vaccine during antenatal care by 5.1 percentage points, a composite antenatal content of care index by 0.16 standard deviations, and HIV testing by 10.4 percentage points [38,41]. Contrary to our study, an equity analysis found that impacts were greatest among the richest, although these results pertain to utilisation rather than quality of care outcomes [42]. In Zambia, P4P was found to have no significant effect on

the availability of inputs (facility infrastructure, drugs, or equipment), process quality of care (antenatal and child health care exit interviews), and client satisfaction [43]. In Cameroon, there was no evidence of an effect of P4P on process quality of care measured using direct observation of antenatal and childcare consultations [44]. Finally, in Benin, P4P had a significant effect on various aspects of clinical care, including a 5.5 percentage point, 2.7 percentage point, and 8.8 percentage point increase in checklists for history taking, physical examinations and advice during antenatal care consultations [45].

Our findings on bonus size connect to a small literature from the US. A study on the Medicare Advantage Quality Bonus Payment Demonstration programme found that doubling the size of payment did not result in better quality of care, possibly because incentives to providers were passed through insurers [20]. A small cohort study found that increasing bonus size by a mean of $3,355 per doctor improved clinical quality of care by 3.2 percentage points [21]. Our study further contributes to this literature by examining changes in quality of care by size of bonus and wealth group, showing that the size of the bonus mattered most for family health teams located in richer areas, but improvement in quality of care in poorer areas was achieved with relatively small bonus payments.

How may bonuses to family health teams have improved quality of care? There are several plausible explanations. First, the bonuses may have increased the motivation of health workers and other family health team members, resulting in greater effort and less shirking. The indicators within the PMAQ score most amenable to worker effort are those concerned with service availability, processes of care, management practices, and possibly utilisation of health care. Second, the channel of influence may have been through family health teams exercising pressure on the health management at the municipality level to improve availability of equipment and medicines. The results from the analysis of the structural quality of care index provide evidence for this second channel. The smaller coefficients, however, also suggest that bonuses may have been more influential in improving indicators of service availability, processes of care, and management practices. Our subgroup results with respect to local area income may be explained by the fact that poorer areas had greater room for improvement (reflecting substantial socioeconomic inequalities in the Brazilian health system) [26,46], hence more potential for the bonuses to bite.

The strengths of the study are the national scale of the analysis, the use of longitudinal data on family health team performance for our quasi-experimental approach, and the availability of a fine-grained measure of local area income with which to examine subgroup effects. Our study has several limitations. First, our study design cannot rule out unmeasured confounding and is therefore unable to provide definitive evidence of a causal relationship between bonuses and improvement in quality of care. It remains possible that the improvement in quality of care in bonus municipalities relative to nonbonus municipalities was due to other time-varying factors that differentially affected the two sets of municipalities. Candidates include political factors, working arrangements, local contracting, existing salaries and other bonuses. Our analysis did, however, control for time-invariant factors at the team level alongside a rich set of municipality characteristics and achieved good baseline balance through the matching procedure. Evidence of a dose–response relationship with respect to bonus size provides further support for a causal interpretation of the findings.

Second, the PMAQ score is not a validated measure of quality. We do not know whether it is a predictor of health outcomes, nor can we be sure it is not vulnerable to gaming on the part of health providers. Whenever measures of performance are linked to financial reward, there is an incentive for gaming. We used the PMAQ score because it is the Brazilian Government's official measure of performance, was developed through a deliberate process with wide consultation, and was based largely on indicators collected independently by universities. Third, our

measure of exposure was based on self-reported data on scheme design collected through an online questionnaire at one point in time. Not only could municipalities have changed how they used PMAQ funds during the study period, recall bias and other sources of measurement error may have affected the reliability of responses, particularly those regarding bonus size. Our large sample size and use of broad bonus size categories will have helped address these issues. Fourth, while we had data on whether bonuses were given by municipalities, we lacked information on how the funds were otherwise used, making a more nuanced interpretation of the findings challenging.

The findings of this study have several implications for policy makers. Because we compared how municipalities used PMAQ funds, while controlling for any differences in the amount of money received, there is no obvious need to assess the cost-effectiveness of bonus payments in the context of our study. Our findings imply that bonuses to family health team workers may be a more effective way of using PMAQ funds than more traditional input-based approaches. However, the important caveat is that we do know whether improvements in the PMAQ score translate into better health and patient experience. Moreover, policymakers must also consider whether the apparent benefit of performance bonuses is likely to fade over time and what the consequences would be for motivation if the bonuses were ever to be withdrawn. Another implication concerns health inequalities in Brazil, which present major health care challenges and were the driver behind introducing PMAQ. While PMAQ was previously found to reduce social inequalities [26], the findings of this study suggest that bonuses may have been a contributor to the overall PMAQ's redistributive effect. It is still worth noting that the short-term gains from P4P can decay in the long run [47]. The Brazilian government has recently rolled out a new primary health care financing scheme, Previne Brasil, to replace PMAQ. It has, however, retained P4P as a central element of the financing mechanism. Findings of this study can help inform design choices going forward.

To conclude, our findings show that giving performance bonuses to staff compared with traditional input-based budgets can potentially lead to a greater improvement in quality of care. This study provides an important contribution to the literature on P4P design with implications for policy makers. Given the wide range of features of P4P design, future research should focus on the effect of other P4P design features, such as payment frequency and recipients as either individual determinants of quality or in combination.

## Supporting information

**S1 Checklist. STROBE checklist completed with section and paragraph numbers for each item.** STROBE, Strengthening the Reporting of Observational Studies in Epidemiology.
(DOCX)

**S1 Portuguese Abstract. Abstract in Portuguese.**
(DOCX)

**S1 Table. Sources of data and variable descriptions.**
(DOCX)

**S2 Table. Probit model used to calculate propensity scores.** The probit regression was run on municipality level data. CI, confidence interval; GDP, gross domestic product; PMAQ, National Programme for Improving Primary Care Access and Quality.
(DOCX)

**S3 Table. Standardised bias before and after matching.** The standardised % bias is the % difference of the sample means in the treated and nontreated (full or matched) subsamples as a

percentage of the square root of the average of the sample variances in the treated and non-treated groups.
(DOCX)

**S4 Table. Income subgroup analysis differences.** The table presents the income subgroup effects as the difference between subgroups (with the poorest group acting as the reference category). Rather than reporting the *p*-value on each subgroup effect, we report the *p*-value from a Wald test that these income subgroup coefficients are jointly equal to zero. CI, confidence interval; PMAQ, National Programme for Improving Primary Care Access and Quality.
(DOCX)

**S5 Table. Difference-in-differences results for structural quality of care.** The dependent variable is the change in the structural quality of care score, which is an index of quality between 0 and 100. The reference groups are as follows: for PMAQ bonus is nonbonus municipalities; for PMAQ bonus size is nonbonus municipalities; for local area is poorest; and for health centre is health post and others. CI, confidence interval; FHT, family health team; GDP, gross domestic product; PMAQ, National Programme for Improving Primary Care Access and Quality.
(DOCX)

**S6 Table. Lagged dependent variable results.** Results are from a lagged dependent variable model based on the full, unmatched, panel of family health teams. The dependent variable is the PMAQ score in round 3. Regressions are at the level of family health teams, with standard errors clustered at the municipality level. The reference groups are as follows: for PMAQ bonus is nonbonus municipalities; for PMAQ bonus size is nonbonus municipalities; for local area is poorest; and for health centre is health post and others. CI, confidence interval; FHT, family health team; GDP, gross domestic product; PMAQ, National Programme for Improving Primary Care Access and Quality.
(DOCX)

**S7 Table. Other robustness checks.** Each panel is a single robustness check, reporting results for the "any bonus" analysis and results for the "size of bonus" analysis. Panel A reports the results from the main analysis in the paper. Panel B is based on the unmatched sample and includes as additional controls: health care spending per capita and whether the political party of the municipality is the same as the national government. Panel C includes additional controls but is based on the matched sample. Panel D uses a smaller calliper of 0.001 in the matching procedure. Panel E uses a larger calliper of 0.2 in the matching procedure. Panel F reports results for size of bonus, including municipalities that gave a variable bonus amount to family health teams. CI, confidence interval.
(DOCX)

**S1 Fig. Histogram of propensity score of treated (bonus) and untreated (nonbonus) municipalities.**
(TIF)

**S2 Fig. Study flow diagram: any bonus.** In the third step, some family health teams had no data on local area income because of missing geographical information to link them to the census area.
(TIF)

**S3 Fig. Study flow diagram: size of bonus.** In the fourth step, some family health teams had no data on local area income because of missing geographical information to link them to the

census area.
(TIF)

**S4 Fig. Violin plot of the PMAQ score by bonus status in the matched sample.** PMAQ, National Programme for Improving Primary Care Access and Quality.
(TIF)

**S5 Fig. Violin plot of the PMAQ score by bonus size in the matched sample.** PMAQ, National Programme for Improving Primary Care Access and Quality.
(TIF)

**S1 Data. Data used to generate Fig 1.**
(XLSX)

**S2 Data. Data used to generate Fig 2.**
(XLSX)

**S3 Data. Data used to generate Fig 3.**
(XLSX)

# Acknowledgments

We thank Allan Nuno Alves de Sousa, Olivia Lucena, Davllyn Anjos, Ilano Barreto, and Wellington Carvalho for their valuable comments and insights throughout the project. We are grateful to the Ministry of Health of Brazil for sharing the data on the PMAQ score and for providing information on the design of PMAQ at the national level.

# Author Contributions

**Conceptualization:** Nasser Fardousi, Everton Nunes da Silva, Josephine Borghi, Timothy Powell-Jackson.

**Data curation:** Nasser Fardousi, Everton Nunes da Silva, Roxanne Kovacs, Letícia Xander Russo, Timothy Powell-Jackson.

**Formal analysis:** Nasser Fardousi, Timothy Powell-Jackson.

**Funding acquisition:** Everton Nunes da Silva, Josephine Borghi, Timothy Powell-Jackson.

**Investigation:** Nasser Fardousi, Timothy Powell-Jackson.

**Methodology:** Nasser Fardousi, Timothy Powell-Jackson.

**Project administration:** Timothy Powell-Jackson.

**Resources:** Timothy Powell-Jackson.

**Supervision:** Timothy Powell-Jackson.

**Visualization:** Nasser Fardousi, Timothy Powell-Jackson.

**Writing – original draft:** Nasser Fardousi, Timothy Powell-Jackson.

**Writing – review & editing:** Nasser Fardousi, Everton Nunes da Silva, Roxanne Kovacs, Josephine Borghi, Jorge O. M. Barreto, Søren Rud Kristensen, Juliana Sampaio, Helena Eri Shimizu, Luciano B. Gomes, Letícia Xander Russo, Garibaldi D. Gurgel, Timothy Powell-Jackson.

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
