## [Editor Report · Decision Letter 0]

9 Dec 2021

Dear Dr Powell-Jackson, 

Thank you for submitting your manuscript entitled "Effect of performance bonuses on the quality of primary care delivered by family health teams in Brazil: A quasi-experimental study" for consideration by PLOS Medicine.

Your manuscript has now been evaluated by the PLOS Medicine editorial staff and I am writing to let you know that we would like to send your submission out for external peer review.

Please re-submit your manuscript within two working days, i.e. by Dec 13 2021 11:59PM.

Kind regards,

Beryne Odeny

PLOS Medicine

---

## [Decision Letter · Decision Letter 1]

15 Feb 2022

Dear Dr. Powell-Jackson,

Thank you very much for submitting your manuscript "Effect of performance bonuses on the quality of primary care delivered by family health teams in Brazil: A quasi-experimental study" (PMEDICINE-D-21-05004R1) for consideration at PLOS Medicine. 

[LINK]

In light of these reviews, I am afraid that we will not be able to accept the manuscript for publication in the journal in its current form, but we would like to consider a revised version that addresses the reviewers' and editors' comments. Obviously we cannot make any decision about publication until we have seen the revised manuscript and your response, and we plan to seek re-review by one or more of the reviewers. 

We expect to receive your revised manuscript by Mar 08 2022 11:59PM. Please email us (plosmedicine@plos.org) if you have any questions or concerns.

We look forward to receiving your revised manuscript. 

Sincerely,

Beryne Odeny, 

PLOS Medicine

plosmedicine.org

1) Please consider identifying the study as a “a difference-in-differences” analysis. Please revise your title according to PLOS Medicine's style. Your title must be nondeclarative and not a question. It should begin with main concept if possible and avoid causal claims such as “Effect” or similar. The study design should remain in the subtitle (i.e., after a colon). Please consider, “Performance bonuses and the quality of primary care delivered by family health teams in Brazil: A difference-in differences analysis”

2) The Data Availability Statement (DAS) requires revision. For each data source used in your study, if the data are not freely available, please describe briefly the ethical, legal, or contractual restriction that prevents you from sharing it. Please also include an appropriate contact (web or email address) for inquiries (this cannot be a study author).

3) Abstract:

a) Please ensure that all numbers presented in the abstract are present and identical to numbers presented in the main manuscript text.

b) Please include the important variables on which the municipalities were matched and any other adjustment variables.

c) In the last sentence of the Abstract Methods and Findings section, please describe the main limitation(s) of the study's methodology.

5) Please conclude the Introduction with a clear description of the study hypothesis.

6) Did your study have a prospective protocol or analysis plan? Please state this (either way) early in the Methods section. 

7) Please clarify terms used such as “supply-side” and “input-based” so that it is clearer for readers who are not familiar with the health context in Brazil.

8) Please ensure that the study is reported according to the STROBE guideline, and include the completed STROBE checklist as Supporting Information. Please add the following statement, or similar, to the Methods: "This study is reported as per the Strengthening the Reporting of Observational Studies in Epidemiology (STROBE) guideline (S1 Checklist)."

9) When completing the STROBE checklist, please use section and paragraph numbers, rather than page numbers.

10) Could you please compare and comment on patient volumes in the bonus vs non-bonus municipalities and how this could be associated with improvement in quality of care? Is this incorporated in the PMAQ score?

11) In the results text and tables, please quantify the main results with both 95% CIs and p values (where applicable).

12) Throughout the text, please remove language that implies causality, such as “effect” or similar. Refer to associations instead.

13) Please do not report P<0.0001; report as P < 0.001.

14) Please define all abbreviations used in tables and figures as footnotes. For example, PMAQ, CI, GDP, FHT

15) References:

a) Please select the PLOS Medicine reference style in your citation manager. In-text reference call outs should be presented as follows noting the absence of spaces within the square brackets: "... countries [1,2]."

b) Please update reference #43 or delete if they have not yet been published

c) Please ensure that journal name abbreviations consistently match those found in the National Center for Biotechnology Information (NCBI) databases. https://journals.plos.org/plosmedicine/s/submission-guidelines#loc-references

d) Ref #34 seems incomplete. Please include journal details or provide a weblink

e) Please provide access dates for all references with weblinks e.g., ref #42

16) Please remove the “Declaration of Interests” and “Funding” statements at the end of the main text. This information is captured in the metadata obtained in the submission form

17) To help us extend the reach of your research, please provide any Twitter handle(s) that would be appropriate to tag, including your own, your coauthors’, your institution, funder, or lab.

Comments from the Academic editor:

The methods and results quite cursory (one reviewer described as generic) so could these be made clearer? For example, what does "allocate funds on basis of performance" mean exactly--what % of the clinic funds was via the PMAQ bonus? A larger concern is about the definition of the outcome measure. What are we measuring exactly using 660 indicators? Which type of indicators predominate--access measures vs quality of care vs volume etc? Reviewer 1 made the excellent point that different indicators went into the index at different time points, potentially rendering the outcome measure incomparable over time. The authors should clarify the extent of variation of the measure. Moreover, a sensitivity analysis that takes a much smaller index indicating quality (versus the other components) that is stable over time would substantially strengthen the analysis

Comments from the reviewers:

Reviewer #1: Thanks for the opportunity to review your manuscript. My role is as a statistical reviewer so my review concentrates on the study design, data, and the analysis. I have put general comments first, and followed these with queries relevant for a specific section of the manuscript (with a page/paragraph reference, page number starting from Abstract).

This study examines whether providing primary care provider team members with a bonus improves quality of care. The quality of care measurement is a composite of many indicators (>200) and used as a continuous variable. The health service areas (municipalities) could decide how to allocate extra funding, either as a bonus to the teams providing care or with centrally (by the municipality) funding of capacity or quality improvement schemes. The design included matching areas who provided bonuses to those that did not, and used a difference-in-difference approach from time-point 2 to the PMAQ score at the time-point 3. 

I was pleased to see that allocation of PMAQ funding at round 1 was included in the propensity score matching - this was an obvious confounder as access to more resources is an obvious way to increase performance/quality. Probit regression was used to estimate the propensity scores - typically, logistic regression gets used here but Probit has potential advantages over logistic regression so this is fine. The analysis includes sensitivity analyses - an unmatched sample, with different matching options (i.e. calliper width), and options for the analysis (i.e. weighted by population and including variable bonuses). OLS is appropriate for the PMAQ as described - with this many indicators it should reliable able to be treated as a continuous variable. The sensitivity analyses were very similar to the main analysis. It is acknowledged that the exposure is based on self-reported data - I would be interested in seeing what the subject-matter reviewers think about potential for measurement error or bias here.

One general issue I have is with the PMAQ. The authors are upfront that this is not a validated measurement of quality of clinical care - this again is one I'd be interested in seeing what the other reviewers think of this. From a data perspective, I am concerned that it seems that the indicators that make up the PMAQ at each time-point could be different. Is there any information that could be added to quantify how many component questions/indicators were dropped or added at each time-point in the study? I would not be concerned if it were only a very few that change, but if it is a relatively large change in composition then effectively it is a different variable at each time-point.

P3, Paragraph 2. So a different set of indicators is collected to make up the PMAQ at each round? 

Would each municipality use the same rules to determine how much bonus each team/team member would receive? I.e. is the bonus a universal scheme for team members? 

P4. Paragraph 1. From the study flow diagrams, it looks like step three was unable to be done for some municipalities. Was this due to gaps in the census data (i.e. data not available for a particular census area?), or were some municipalities unable to be allocated to a census area? 

P5, Paragraph 1. What method was used to adjust the SEs for clustering at the municipality level? 

P5, Paragraph 3. The direct comparison is ok, but I would also consider adding standardised differences for the variables before/after matching as well as this make the comparison easier. The other check that would be useful to see would be something to demonstrate the degree of common support between the two groups. The last time I used PS matching in Stata (a long time ago though!) the add-on psmatch2 had some basic visualisations for distribution of the propensity score (i.e. an overlapping density plot by group on the same graph) which were effective that should be included in the appendix for the main analysis. There might even be something prettier now.

P5, Paragraph 4. The stratified analysis approach will work - but usually the better way to do this is to include an interaction of the treatment effect with the sub-group variables in the regression model and use an appropriate command/syntax (i.e. margins in Stata) to get treatment effect estimates for each sub-group combination. If only the key variables (i.e. time, treatment group) are included then a stratified approach will effectively the same estimate. If you have included the variables from the propensity score matching process into the final D-in-D model (i.e the 'double-robust approach') then doing a stratified analysis is effectively like including an interaction term of the sub-groups with these variables as well which may not be what was intended. The benefit of the interaction approach is that you can get a p-value for the interaction indicating the level of evidence there is for a difference in treatment effect according to the sub-groups (a more direct approach than comparing 95% CIs which is an approach that is less powerful than the p-value from the interaction). 

P5. Paragraph 5. I am clear about sensitivity analyses except for weighting by population - is this to get results that are interpretable at a population level?

P6. Table 1. I wasn't clear if the matching was 1:1 and without replacement, how the matched sample ended up with more teams who received any bonus than those with no bonus? i.e. shouldn't it be 5052 teams in both groups in the matched sample?

Supp appendices.

Fig 1. I'd recommend to use an alternative plot to the bar plot with CI lines - depending on the size of the dataset in each of the bonus categories, a boxplot with jittered points, or if too much data then a plot that shows the distribution would be much more useful, i.e. a 'violin plot'.

Fig 2. I would clarify this as the 'income of team members' in the caption.

Reviewer #2: 

Thank you for sharing this interesting paper on a major P4P programme in Brazil. The work is very well written and expertly carried out. The writing is clear and understandable, and the methods are appropriate. The inclusion of multiple sensitivity analyses is welcome and these demonstrate the robustness of the work. I have no major comments or changes suggested.

A few very minor comments:

- Clarify early on (in abstract) that this study is about evaluating rewarding health professionals, as the term "family health teams" could imply service providers or their budgets. 

- I think it would be useful to mention the size of the overall PMAQ as proportion of public spending - I believe it was relatively small. 

- Would it be worth mentioning in the limitations that municipalities that give incentives to teams could be different in characteristics that were not measured (political factors, working arrangements, local contracting, existing salaries and other bonuses) and could have had some minor bias;

- I would emphasize more the uncertainty about clinical relevance of the PMAQ surveys and scores; 

- Could the authors also comment on the size of the bonus - 21% or 50%+ of salary - where most of the impact was found. I think this is a relatively large amount, and do they recommend this approach be used in other settings for relatively modest quality gains?

Reviewer #3: Paper studies "a national health financing programme to improve access to and quality of primary health care" (PMAQ) in Brazil. Through this program, "the federal government made financial payments to municipalities based on the performance of family health teams. […] Municipalities had the flexibility to decide whether to retain payments at the municipal level or redirect them to the family health team level, […] could decide how to use the financial resources which means we can compare municipalities that gave bonuses to health workers with those that chose to invest solely in the supply-side readiness of the primary care system. Finally, municipalities differed in the size of bonus paid to family health teams."

From this initial description, the reviewer is sceptical about the authors claim that "PMAQ provides an ideal testing ground for addressing three questions about scheme design …," because much of the design features are the consequence of decisions taken at the municipal level, the classification as a "quasi-experimental" study is therefore misleading, as is the statement that the study "exploited municipality variation in the design features" of the national program. Instead, the design of PMAQ appears uniform across municipalities; within this framework, the study addresses choices made to best effect by municipal authorities.

The study adopts a difference-in-difference approach, comparing changes in outcomes (a composite indicator on health service provision collected under PMAQ) across municipalities adopting bonuses vs those who spent it on supply-side strengthening. Potential bias from self-selection of municipal authorities into arrangements on how the funds are used is addressed by a procedure matching each of the "bonus" municipalities to a control. The methods and their motivation are explained adequately though somewhat generically. Some important aspects of the methods are left unclear: (1) What is the "full set of baseline municipality, facility and local area characteristics" the study controls for? (2) The matching procedure, a crucial aspect of the study as it potentially mitigates the bias inherent arising from the endogeneity of the design decisions on the municipal level, is documented rather parsimoniously (considering it is crucial in relation to the objectives of the study), largely indirectly in connection with the data description under results (Table 1). (3) Relatedly, the final step in the process by which the sample is whittled down to 2346 municipalities through the matching procedure is opaque.

Result tables are presented and summarized competently. However, two figures referred to in the text are missing from the paper.

The discussion of shortcomings of the study is competent, making pointers to the issues raised in this review. E.g., it notes that "our study design cannot rule out unmeasured confounding and is therefore unable to provide definitive evidence of a causal relationship."

Overall, this is a competent study. The style and presentation are crisp and language is consistently of high standard. However, the framing of the study as an ideal setting for a DiD approach is not supported by the data characteristics - the fact that the decisions on the design features the study focuses on are made on the municipal level is highly problematic for the purposes of the analysis. This - and the sensible steps the authors have taken to mitigate this problem - needs to be communicated more clearly.

Editorial note: Consider adding a digit to some estimated coefficients which are small in absolute numbers (e.g., "0.0" for population size or clinical staff).

Reviewer #4: This is a very nice written report on assessment of a large pay-for-performance (P4P) programme (called PMAQ) in Brazil Federal 5000 municipalities to improve primary health care quality. The authros used a quasi-experimental design differences-in-differes with matching (and without matching). They took leverage fact that the money was sent to the municipal level which decided how to use (from improving infrastructure to incentivise the health workers/health workers family teams). These actions where quite variable to build different forms of the exposure variable: either binary (any bonus to family health teams vs none) and multinominal (levels of the size of bonus). They found statistically significant association (apparentely U-shaped) robust to many sensitivity analyses.

Few issues:

 1. It would be good to have further details on how this PMAQ score (which varies from 0 to 100) is built. 

- Please add further details on this.

- The authors use linear regression to analyse the change of the PMAQ score (as continuous variable) however none knows what that means in substantive terms (the validity issue the authors point out in the limitation) (eg: is a change of 1 the same as in 10 unities). This makes quite hard to appreciate in substantive terms the statistically significant results presented. 

- Please provide more descriptive statistics of PMAQ round 1 (eg median and IQR on table 1) and as well as for PMAQ 3 and the change of the score. Had you done deciles of PMAQ round 1 how many would improved their deciles for example? This is OK to be as supplementary materials. 

 2. Table 1: the total population in what unities is this? 

 3. Table 2 and similar tables in the supplementary materials: please add the reference category for the binary and categorical variables (PMAQ bonus, PMAQ bonus size, Local area and health facility type)

[LINK]

---

## [Decision Letter · Decision Letter 2]

13 Apr 2022

Dear Dr. Powell-Jackson,

Thank you very much for re-submitting your manuscript "Performance bonuses and the quality of primary health care delivered by family health teams in Brazil: A difference-in differences analysis" (PMEDICINE-D-21-05004R2) for review by PLOS Medicine.

I have discussed the paper with my colleagues and it was also seen again by two reviewers. I am pleased to say that provided the remaining editorial and production issues are dealt with we are planning to accept the paper for publication in the journal.

[LINK]

We look forward to receiving the revised manuscript by Apr 20 2022 11:59PM.   

Sincerely,

Beryne Odeny, 

PLOS Medicine

plosmedicine.org

Requests from Editors:

1) Please include line numbers in your next draft

2) In the abstract and main text please avoid implications of causality (“... increase in the PMAQ score”; “positive impact”, “a more effective way”); however, we kindly request that you use more restrained language in describing findings, e.g. “associations” ,“evidence of an apparent benefit …”or similar

3) Regarding data availability, thank you for providing an appropriate contact for inquiries on restricted data. This will suffice for researchers who may wish to obtain the full data set for PMAQ score variables and exposure variables to replicate analyses. 

4) For the restricted data, please provide a “minimal data set” which consists of the data set used to reach the conclusions drawn in the manuscript with related metadata and methods, and any additional data required to replicate the reported study findings in their entirety. Authors do not need to submit their entire data set, or the raw data collected during an investigation. Please submit the following data:

a) The values behind the means, standard deviations and other measures reported;

b) The values used to build graphs;

c) The points extracted from images for analysis

5) Introduction - first sentence of the last paragraph should read “mattered in Brazil,” and not “mattered. in Brazil.”

6) In your tables (e.g. S6, S7) please do not report P<0.0001; report as P < 0.001

7) References – ref #38 seems incomplete. Please include the journal details

8) Please remove the “Data availability” statement at the end of the main text. This information is captured in the metadata obtained in the submission form

Comments from Reviewers:

Reviewer #1: hanks for the revised manuscript and replies to my initial queries. Overall the manuscript looks good to me and I recommend it should be published subject to a few small adjustments (at the end). The limitations of the PMAQ are made clear - I don't have any objections here and it looks like overall the other reviewers are satisfied with using this as a measure of quality of care. Perhaps this points to the need for someone to develop a QoC index for LMICs? 

The description of PMAQ is much clearer. Sensitivity analysis (with reduced set) provides reassurance, they are smaller but I agree that because the same score is applied across all sites/teams (and D-I-D is used) that it would take an unusual mechanism for a charge in the components of the score to lead to a biased estimate of effect. 

The standardised difference in the supplementary table and Fig S1 are useful diagnostic information for the review - these look fine to me. 

I don't see any major problems with exclusion of FHTs - it looks as though the number excluded through this process was consistent between the bonus/non-bonus strata. 

Abstract, Methods and finding: I'd say '…least square regression' instead of the plural

For the sub-group analyses, to clarify, the p-value I referred to in the earlier review was the overall test of including the interaction - you could do this with lrtest "interaction*term" in Stata that provides an overall test the model with the interaction included vs. a model without it. The individual p-values for each marginal effect aren't really needed, and you could just remove these and add the overall p-value for each sub-group section (e.g. 1 p-value for PMAQ bonus). 

Reviewer #3: I appreciate the diligent responses to the comments provided by this reviewer and others.

The paper overall reads much better now, and the analysis has become more transparent.

Some of my comments have been addressed either directly (e.g., R3.2), or become redundant following clarification of the matching procedure (e.g., R3.3, R3.4). The paper now is much clearer in methods and limitations (R3.7), also in response to the constructive comments provided by Reviewer 1.

Having looked at the responses to comments specifically, I also leaned back and attempted to re-read the paper with a fresh eye. While I described it as an overall competent paper in the previous round, it has improved in terms of transparency and precision on methods and interpretation of results. I now find it ready to go, and have no further observations which need to be raised at this point.

[LINK]

---

## [Decision Letter · Decision Letter 3]

10 May 2022

Dear Dr. Powell-Jackson,

Thank you very much for re-submitting your manuscript "Performance bonuses and the quality of primary health care delivered by family health teams in Brazil: A difference-in differences analysis" (PMEDICINE-D-21-05004R3) for review by PLOS Medicine.

I have discussed the paper with my colleagues and the academic editor and it was also seen again by one reviewer. I am pleased to say that provided the remaining editorial and production issues are dealt with we are planning to accept the paper for publication in the journal.

[LINK]

We look forward to receiving the revised manuscript by May 17 2022 11:59PM.   

Sincerely,

Beryne Odeny, 

PLOS Medicine

plosmedicine.org

Requests from Editors:

1) Discussion line #347, should read “… overall impact on PMAQ” and not “…overall impact of PMAQ)

2) Discussion line # 387 is missing a word. It should read “..the availability of a..”

Comments from Reviewers:

Reviewer #1: I think there's just one adjust to make, and it is regarding the p-values for each marginal effect. I would agree that overall there are general consistent effects across the subgroups, the issue is that comparing p-values between levels of a sub-group variable is a phenomenon called 'differences in nominal significance', where the bonus * subgroup effect is checked by comparing the whether the p-value for one subgroup is above/below the threshold vs. another p-value. There are good reasons not to do this (it often leads to the wrong inference, e.g. in an extreme situation where one subgroup level has p=0.051 and another p=0.049). The most powerful approach to testing whether or not there are consistent effects of payment according to PMAQ/municipality characteristics is a likelihood-ratio test of all parameters associated with the interaction between bonus and subgroup. For example, with a simple model (assuming simple dummy coding of area SES level and bonus status):

reg pmaq bonus_status area_ses bonus_status*area_ses

we would check whether there is overall heterogeneity in the effect of bonus with a post-estimation command:

lrtest bonus_status*area_ses

(it has been a while since I've used Stata so the code might not be 100% accurate). For each subgroup variable this p-value can replace all individual p-values of marginal effects, and allows a direct test of heterogeneity.

The manuscript looks great otherwise and I've recommended it should be accepted subject to this one last update.

[LINK]

---

## [Decision Letter · Decision Letter 4]

26 May 2022

Dear Dr Powell-Jackson, 

On behalf of my colleagues and the Academic Editor, Dr. Margaret Kruk, I am pleased to inform you that we have agreed to publish your manuscript "Performance bonuses and the quality of primary health care delivered by family health teams in Brazil: A difference-in differences analysis" (PMEDICINE-D-21-05004R4) in PLOS Medicine.

PRESS

Sincerely, 

Beryne Odeny 

PLOS Medicine